# Prevalence and Risk Factors for ESBL/AmpC-*E. coli* in Pre-Weaned Dairy Calves on Dairy Farms in Germany

**DOI:** 10.3390/microorganisms9102135

**Published:** 2021-10-12

**Authors:** Laura Patricia Weber, Sylvia Dreyer, Maike Heppelmann, Katharina Schaufler, Timo Homeier-Bachmann, Lisa Bachmann

**Affiliations:** 1Alta Deutschland GmbH, 29525 Uelzen, Germany; weber.laura@fbn-dummerstorf.de; 2Clinic for Cattle, University of Veterinary Medicine Hannover, Foundation, 30173 Hannover, Germany; maike.heppelmann@tiho-hannover.de; 3Research-Institute for Farm Animal Biology (FBN), Institute of Nutritional Physiology “Oskar Kellner”, Wilhelm-Stahl-Allee 2, 18196 Dummerstorf, Germany; 4Friedrich-Loeffler-Institut, Institute of International Animal Health/One Health, Greifswald—Insel Riems, 17493 Greifswald, Germany; sylvia.dreyer@fli.de; 5Institute of Pharmacy, Universität Greifswald, 17475 Greifwald, Germany; katharina.schaufler@uni-greifswald.de; 6Institute of Infection Medicine, Christian-Albrecht University and University Medical Center Schleswig-Holstein, 24118 Kiel, Germany; 7Friedrich-Loeffler-Institut, Institute of Epidemiology, Greifswald—Insel Riems, 17493 Greifswald, Germany; timo.homeier@fli.de; 8Faculty of Agriculture and Food Science, University of Applied Science Neubrandenburg, Brodaer Str. 2, 17033 Neubrandenburg, Germany

**Keywords:** ESBL/AmpC-*E. coli*, calves, cattle, antimicrobial resistance

## Abstract

The objectives of this study were to ascertain the fecal ESBL/AmpC-*E. coli* prevalence and to detect risk factors for their occurrence in young pre-weaned calves and their dams on large dairy farms in Germany. From 2018–2019 we investigated 2816 individual fecal samples from pre-weaned dairy calves and their dams, representing seventy-two farms (mean = 667 milking cows) from eight German federal states. To assess possible risk factors associated with ESBL/AmpC-*E. coli* prevalence in calves and dams, a questionnaire was performed, collecting management data. We observed an ESBL/AmpC-*E. coli* prevalence of 63.5% (95% CI: 57.4–69.5) among the sampled calves and 18.0% (95% CI: 12.5–23.5) among the dams. On all farms, at least one positive sample was obtained. To date, this is the highest ESBL/AmpC-*E. coli* prevalence observed in dairy herds in Europe. Feeding with waste milk was identified as a significant risk factor for a high prevalence of ESBL/AmpC-*E. coli* in calves. Many calves at large dairies in Germany are fed with waste milk due to the large amounts generated as a result of antibiotic dry-off routines and mastitis treatment with antibiotics. Other notable risk factors for high ESBL/AmpC-*E. coli* in calves were the general fitness/health of dams and calves, and the quality of farm hygiene. Taken together, these findings suggest that new or improved approaches to animal health management, for example, antibiotic dry cow management (selective dry cow therapy) and mastitis treatment (high self-recovery), as well as farm hygiene, should be researched and implemented.

## 1. Introduction

The World Health Organization (WHO) has acknowledged antimicrobial resistance as one of today’s biggest threats to global health, food security and development. In 2015, they launched the *Global Action Plan on Antimicrobial Resistance* (GAP AMR). The GAP AMR clearly specifies that AMR affects sectors beyond human health, and that therefore, a One Health approach is essential in order to control the spread of AMR [1].

Among the most critical antibiotic resistant bacteria, extended-spectrum β-lactamases (ESBL)-producing Enterobacteriaceae are listed on the WHO’s *Global priority list of antibiotic-resistant bacteria to guide research, discovery and development of new antibiotics* [2]. ESBL-*E. coli* in particular show high zoonotic potential and belong to the most important multi-resistant pathogens worldwide. Resistance to broad-spectrum β-lactams can be mediated by ESBLs and AmpC β-lactamases (AmpC). Both can hydrolyze 3rd- and 4th-generation cephalosporins such as cefotaxime and can either be found separately or in coexistence [3,4].

Enterobacteriaceae carrying ESBL, especially *Escherichia* (*E.*) *coli*, are a public health challenge for cattle production across the world [5]. ESBL-*E. coli* isolates are frequently reported in samples of livestock origin (e.g., dairy cattle) and associated food products, increasing the potential for human transmission [6,7]. Although commensal *E. coli* rarely cause infections, they may horizontally transfer resistance genes to pathogenic bacteria, thereby contributing to the development of severe infections and the spread of AMR [8,9]. [2]. Although *E. coli* from cattle feces carry less antibiotic resistance than feces from other livestock animals, young calves have frequently been proven to carry ESBL-*E. coli* [10,11]. A U.S. study by Mir et al., found colonization with cefotaxime-resistant bacteria (predominantly *E. coli*) in more than 92% of young beef calves sampled in their study. Notably, the investigated calves had never been treated with antibiotics, suggesting acquisition from another source [12]. One study showed a link between feeding with antibiotic-containing milk and the presence of ESBL-*E. coli* in calves. Although this link could not be confirmed by other studies [13,14,15], it is consistent with the finding that the management of dairy farms appears to promote the carriage of ESBL-*E. coli* in cattle when compared to farms focusing on beef production alone [16].

In Germany, a number of studies have already investigated the prevalence of ESBL-*E. coli* in dairy farms. The results of those studies revealed a high ESBL-*E. coli* prevalence; however, they were limited in that they only investigated dairy products, small herds, or low sample sizes [6,7,10,16,17,18]. To address these limitations, the present study aimed to analyze the prevalence of ESBL-*E. coli* on a larger scale in large German dairies. Here, we investigated both the prevalence of ESBL/AmpC-*E. coli* in the feces of calves and their dams, and how that correlated with the ESBL/AmpC-*E. coli* load in the associated farm milk bulk tanks. In addition, we analyzed herd management data to determine the accompanying risk factors.

## 2. Materials and Methods

### 2.1. Study Design

A cross-sectional study design was used to determine the prevalence of ESBL/AmpC-*E. coli*-positive fecal samples of calves and cows in German dairy farms from December 2018 until August 2019. In addition, potential risk factors associated with high ESBL/AmpC-*E. coli* prevalence were analyzed.

### 2.2. Participation Criteria

Only dairy farms listed in the client-base of Alta Deutschland GmbH (Uelzen, Germany) were included. The farms either participated in the Alta benchmarking program (a program providing farmers with information comparing the performance of their own farm to other participants in the program on various operational parameters) or operated a herd management program. Potential participants were referred to the study by the company’s field staff and asked to participate. Beyond allowing the sampling of their animals, participants also agreed to provide the study with access to data from the herd management program, and to any veterinary documents recording pharmaceutical applications and/or deliveries over the last six months. All data were collected and processed in strictly pseudonymous form, including blinded examination of the samples in the lab.

### 2.3. Sampling and Transportation

Fecal swabs were collected from calves between seven and 28 days of age and their corresponding dams. On the day of sampling, the required data for all sampled calves were extracted from the herd management program. The sample size was calculated using the https://epitools.ausvet.com.au/oneproportion (last accessed on 31 August 2019) sample size calculation tool, and was based on the number of appropriately aged calves present on the day of sampling. We chose to use an expected prevalence of 10%, a confidence interval of 95%, and a desired precision of 5% [15]. Only dams were sampled of which the calves were also sampled. In Appendix A, we provide a table summarizing the number of calves per farm and the number of calves sampled per farm. Calves were selected using a pseudo-random strategy to allow for an attempt at achieving an equal distribution of male and female calves and younger and older calves. The exact assignation of calves to their dams was possible based on the living ear tag ID of each animal registered in the herd management program. Fecal swab samples (Sigma Transwap, MWE, Wiltshire, UK) were stored in Amies medium at 5 °C.

In addition to collecting fecal samples, milk from the collection tank of each farm was also sampled using a 50 mL Falcon tube (*n* = 72). Seventy-two dairy farms participated in the study. The farms housed between 134 and 1697 (mean = 667) dairy cows and were investigated between December 2018 and August 2019. A total of 2816 animals were sampled; 1442 of them were calves and 1374 were dams. Since male calves usually leave farms at around 14 days of age, the number of female samples predominated (female = 68%, male = 32%). Due to twin births and dam deaths, fewer dams than calves were sampled.

### 2.4. Location of the Farms

To illustrate the location of farms that participated in the study, a map was created using Kartenexplorer Version 2.04 (R6) (http://kartenexplorer.fli.de/ (accessed on 20 February 2021)). The farms were mainly located in the north and east of Germany (Figure 1).

### 2.5. Questionnaire and Data Collection

Together with the herd manager, a questionnaire was completed. The questionnaire collected information regarding potential risk or protective factors for the prevalence, or transmission and distribution, of ESBL/AmpC-*E. coli* within the farm. Furthermore, farm-specific indicators from the herd management programs and, if applicable, the Alta benchmarking-data, were compiled, extracted and analyzed. The veterinary documents recording all pharmaceutical treatments that occurred in the six months prior to sampling were reviewed, and the classes of antibiotic agents used were noted. The complete questionnaire is provided in Appendix A.

### 2.6. ESBL/AmpC-*E. coli* Isolation and Characterization

Fecal and milk samples were cultured on CHROM ID agar plates (Mast Group, Reinfeld, Germany). Merlino et al. showed that CHROM ID agar plates are particularly suitable for the identification of *E. coli* due to their high specificity [19]. Furthermore, 2 µg/mL cefotaxime (Alfa Aesar by Thermo Fisher Scientific, Kandel, Germany) was added to the agar plates, as Vinueza-Burgos et al. demonstrated that supplementation with cefotaxime led to the identification of ESBL/AmpC-carrying *E. coli* with high specificity. In a study of cefotaxime-resistant *E. coli* from broiler farms, an ESBL/AmpC phenotype (or even an MDR phenotype) was detected in 98.3% of cefotaxime-resistant *E. coli* [20]. Agar plates containing cefotaxime were incubated overnight at 37 °C.

According to the manufacturer’s protocol, pink-violet colored, shiny colonies represent ESBL/AmpC-*E. coli*-positive results. Positive colonies were picked and sub-cultivated on CHROM ID agar plates supplemented with Cefotaxim until a pure culture was achieved. Considering the high number of isolates obtained in this study and the focus of the study (i.e., identifying risk factors for a high prevalence of resistant pathogens in feces in dairy farms), further differentiation of isolates was not performed.

### 2.7. Data Analysis

For statistical analyses, we used SAS (version 9.4, SAS Institute Inc., Cary, NC, USA) and the glm function of the stats package (R Core Team (2019) in R: A Language and Environment for Statistical Computing (R Foundation for Statistical Computing, Vienna, Austria. Available online: https://www.R-project.org/ (accessed on 1 March 2021)) with the argument family = ‘binomial’. Logistic regression was applied to analyze the influence of the binomial parameters on the prevalence of ESBL/AmpC-*E. coli*. We classified farms as having a high or low prevalence in cows according to Gonggrijp et al. (2016). Thus, farms with 0% maternal prevalence were classified as low-prevalence farms, whereas farms with at least one or more positive dams were classified as high-prevalence farms. However, the Gonggrijp et al. (2016) classification system was not appropriate for categorizing the farms as having a high or low prevalence with respect to ESBL/AmpC-*E. coli* in calves, as the overall prevalence was higher in this group. Therefore, farms with a calf prevalence <50% were classified as low-prevalence and >50% classified as high-prevalence; this cut-off value resulted in nearly the same number of farms in the low category for calves as in the low category for the cows. To build the multivariable model, we first screened all the parameters derived from the questionnaire in a univariable logistic regression model and only those parameters that demonstrated an association with ESBL/AmpC-*E. coli* prevalence at *p*-values ≤ 0.15 were then entered into the multivariable model. With a forward selection and elimination procedure at each run, the variable with the lowest *p*-value in the univariable analysis entered the model. Once all variables had *p*-values ≤ 0.05 and the lowest Akaike information criterion value was reached, the final model was defined [21]. Potential confounders were explored by monitoring for change in the coefficient of a parameter after removing another one. The R package “Interaction” was used to exclude potential interactions between the parameters. The coefficient of determination (pseudo-R^2^) was calculated to measure the proportion of variance explained by the model.

To check for associations between all metric farm data collected in the questionnaires, and the level of ESBL/AmpC-*E. coli* prevalence in calves and dams, correlation analysis was applied. Linear correlation between the variables was measured using Pearson correlation coefficient, whereas rank correlation was determined using Spearman’s rank correlation coefficient. *p*-values ≤ 0.05 were considered significant.

## 3. Results

### 3.1. ESBL/AmpC-*E. coli* Prevalence of Calves and Cows

Descriptive statistics of ESBL/AmpC-*E. coli* prevalence are presented in Table 1. The results revealed that the mean farm ESBL/AmpC-*E. coli* prevalence in calves was almost 3.5 times higher (mean = 63.5%) than that of the dams (mean = 18.0%). Nearly 14% of the calf-cow pairs were positive for ESBL/AmpC-*E. coli*. In all farms, at least one positive sample was obtained.

All bulk milk tank samples were negative for ESBL/AmpC-*E. coli.*

### 3.2. Description of Farm Management Practices

Analysis of the questionnaire revealed that the application of ß-lactam antibiotics was a common management practice among all participating farms, followed by antibiotic dry cow therapy in general (91.7%). Disinfection of the calving area was only applied in 13.9% of the farms and 55.6% of the farms used chinolons. In 50.0% of the farms, every case of clinical mastitis was treated with antibiotics (Table 2). All the results of the questionnaire are provided in Appendix A.

The analysis of calf-related feeding factors showed that feeding with colostrum until three hours after birth was practiced on 81.9% of the farms (Table 3). Nearly half of the farms fed four or more liters of colostrum. Following colostrum feeding, 61.1% of the calves received 5 L or less milk or milk replacer (MR) per day. In 66.7% of the farms, calf rations included waste milk (WM, non-saleable/discard milk, colostrum/milk of cows treated either with antimicrobials or drugs), whereas MR was exclusively fed on 19.4% of the farms.

### 3.3. Risk Factors for the Occurrence of ESBL/AmpC-*E. coli* in Calves and Cows

The results of the univariable logistic regression model (*p* < 0.15, Wald test) analyzing the associations between the farm management parameters described above and a farm being classified as high ESBL/AmpC-*E. coli* prevalence in calves are listed in Table 4.

Based on the results of the multivariable logistic regression analysis, the final model contained three parameters and explained 31% of the variation in ESBL/AmpC prevalence (pseudo-R^2^ = 0.31). Feeding WM to calves showed the strongest prevalence-increasing effect with respect to ESBL/AmpC-*E. coli* in calves (OR = 9.65; *p* = 0.005), whereas receiving a preventive treatment for cows had a significant ESBL/AmpC-E. *coli* prevalence-decreasing effect (OR = 0.13; *p* = 0.029) (Table 5). In addition, the implementation of daily cleaning of the calf feeding equipment (OR = 6.03; *p* = 0.021) increased the calf prevalence of ESBL/AmpC-*E. coli* as well (Table 5).

The results of the association between the potential risk factors and the maternal ESBL/AmpC-*E. coli* prevalence according to the univariable logistic regression model are listed in Table 6. Based on the results of the univariable logistic regression model for the ESBL/AmpC-*E. coli* prevalence in dams, no final multivariable logistic regression model could be created. In this respect, the results of the univariable analysis (Table 6) were used for further interpretation.

The results of the linear correlation showed a significantly negative relationship (*r* = −0.26, *p* = 0.04) between the actual lifetime production of the dairy herd and the ESBL/AmpC-*E. coli* prevalence of the dams. The results of the monotonic correlation showed a distinctively positive relationship (*ρ* = 0.23, *p* = 0.06) between the culling rate of the dairy herd (i.e., the percentage of animals that leave the herd for different reasons (disease, infertility, low milk yield, etc.)) and the ESBL/AmpC-*E. coli* prevalence of the dams. For all other metric data collected, no correlation was found, e.g., calf prevalence did not correlate with maternal prevalence or farm size.

## 4. Discussion

### 4.1. Prevalence Data

With 72 farms from eight different German federal states, comprising 2816 individual samples of pre-weaned dairy calves and their dams, our study represents the largest study on the fecal ESBL/AmpC-*E. coli* prevalence in German dairy farms to date. The results provide evidence for the occurrence of ESBL/AmpC-*E. coli* in 100% of the examined dairy farms. The herd prevalence was higher than that of all previous European prevalence studies [11,13,14,15,16,22,23,24]. However, in those studies the dairy farms investigated were all, on average, considerably smaller in size and lower in the milk production performance level compared to the farms included in the present study. In contrast to other studies, our study omitted the enrichment of the sample material prior to cultural investigations. Omitting the enrichment step in our study could lead to an underestimation of prevalence. Therefore, our result that in 100% of the herds at least one animal tested positive for ESBL/AmpC-*E. coli* is concerning. Only one other study, a Swedish study, used the same approach and determined a herd prevalence of 18% [15]. Until now, the study with the highest ESBL/AmpC-*E. coli* herd prevalence in European dairy herds (93%) was also carried out in southern Germany [13]. In Germany, there is a north–south divide in cow numbers on dairy farms. In the north of Germany, there are larger farms with more than 100 cows, whereas in the south the herds are smaller. This is the first study with individual samples in large dairy herds in Germany. Due to convenient sampling, there could be some bias regarding the estimation of prevalence in our study. However, in a quarter of the herds (*n* = 18) all calves present were sampled (Appendix A).

In the present study, the number of ESBL/AmpC-*E. coli*-positive calves per farm varied between a single animal and all sampled animals, with an average calf prevalence of 63.5%. This enormous range in variation has been observed previously in other studies using a similar methodology [15,25,26]. Although much lower on average (18.0%), prevalence variations in dams also showed a wide range (0% and 88.9%). This finding was consistent with a Czech study from Manga et al. (2019) where a maternal prevalence of 15% was determined. However, that study only investigated a single dairy farm [26]. Other studies differed in the methodology used, which allows no comparison with the work described here [13,16,27,28,29]. Further studies are needed to determine whether the ESBL/AmpC-*E. coli* colonization of the calf originates from the dam or from other sources.

### 4.2. Risk Factors for the Occurrence of ESBLAmpC-*E. coli* in Calves

The most important risk factor associated with high prevalence of ESBL/AmpC-*E. coli* in calves was the feeding of WM, whereas the use of MR was associated with a low ESBL/AmpC-*E. coli* prevalence. Although prohibited for human consumption, feeding of WM to calves is a common practice worldwide [30,31], as it is both legal and economically beneficial. In general, WM often contains antibiotic residues, albeit far below the therapeutically effective dose [32,33]. It is known that feeding milk containing even very low concentrations of antibiotic residues can provide a selection advantage for multidrug-resistant pathogens [34]. In a study by Pereira and colleagues, resistant bacteria were found in a significantly greater proportion of calves which were fed with antibiotic-spiked milk compared to calves, which were fed raw milk without antimicrobials. Moreover, they detected a clear difference in the calves’ microbial colonization profiles [35]. Additionally, in 2014, Brunton et al. detected a higher amount of ESBL-*E. coli* in the feces of calves fed with WM, as well as a prolonged excretion of the multidrug-resistant pathogens after weaning, and considered a selective effect of WM, although it is unclear at what concentrations the antibiotic residues of WM reach the intestines of the suckling calves [32]. Likewise, contamination of WM with ESBL-*E. coli* has been shown to occur, which could also lead to oral colonization of suckling calves [36]. However, if the intake of vital ESBL-*E. coli* via WM was causal for an increased prevalence in calves, pasteurization should be a protective factor, which could not be proven by several studies [37,38].

Our finding that feeding MR was associated with a low ESBL/AmpC-*E. coli* prevalence was consistent with a Swedish study by Duse and colleagues in 2015, in which calves fed with MR excreted significantly less ESBL-*E. coli* than those fed with raw milk [15].

Another parameter significantly associated with a farm being classified as having a high ESBL/AmpC-*E. coli* prevalence in calves was daily cleaning of the calf-feeding equipment, such as nursing buckets. This result is supported by a study by Heinemann and colleagues (2020), which showed that current hygiene practices in dairy farms are often inadequately implemented. In that study, the highest bacterial loads, including those of ESBL-*E. coli*, were detected in the milk-feeding buckets and inner artificial teats, although sanitation measures were in place [39]. It is conceivable that insufficient cleaning leads to a transmission of bacterial colonization between calves, especially when nursing buckets are not permanently allocated to the same calf. Another possible explanation is that the feeding equipment is contaminated through the use of a single batch of cleaning solution for all equipment, promoting a microbial transmission. Although this theory of the bacterial contamination of feeding equipment is as yet unproven, bacterially contaminated teat cleaning solutions have been identified as a cause of mastitis [40].

The survey conducted among the farmers suggests an association between high ESBL/AmpC-*E. coli* colonization of the calves and the general application of preventive treatment to dairy cows on the farms. However, due to the wide range of possible prophylactic actions taken between the participants (e.g., milk fever/ketosis/acidosis prophylaxis), interpretation of this parameter is difficult. Nevertheless, regardless of the exact treatment regimen, disease prevention measures appear to improve maternal fitness and thereby lead to a decrease in ESBL/AmpC *E. coli* prevalence. One possible explanation for this association is that farms which practice the general application of preventative treatments also have a generally higher quality of farm management and therefore cows with overall greater fitness. It is assumed that cows with a greater fitness are less colonized with ESBL/AmpC-*E. coli*, which could result in a lower infection risk for the newborn calves at the time of birth. Moreover, cows from well managed farms are potentially in a better constitution for parturition and are monitored better during birth. This might result in easier births and subsequently fitter and healthier calves, given that difficult births are known to weaken calves [41]. It is also conceivable that cows with a high fitness respond immunologically well to their environment and secrete high-quality colostrum. This in turn has a positive effect on the newborn calves and makes them more resistant to colonization with ESBL/AmpC-*E. coli* present in their environment [42].

### 4.3. Risk Factors for the Occurrence of ESBLAmpC-*E. coli* in Cows

Another parameter aiming to achieve a better fitness is the result of the linear relationship showing a significantly negative correlation between the actual lifetime production of the dairy herd and the maternal ESBL/AmpC-*E. coli* prevalence. On the one hand, this observation may point to better hygiene, housing, feeding and health management within the dairy herds, leading to greater general health, well-being and fitness of the cows, resulting in a lower susceptibility to ESBL/AmpC-*E. coli* [43]. On the other hand, a possible explanation for the lower prevalence of ESBL/AmpC-*E. coli* in dams on farms with higher actual lifetime production could be the lower use of antibiotic therapeutics within these farms due to a generally better physical constitution of these animals. The positive relationship between the culling rates and ESBL/AmpC-*E. coli* prevalence of the dams supports the association between better health and the lower occurrence of multi-resistant bacteria.

Farms that practiced dry teat cleaning before milking showed significantly lower ESBL/AmpC-*E. coli* prevalence in the examined dams. A prerequisite for dry teat cleaning is good barn hygiene. Therefore, a possible explanation for the lower maternal prevalence in herds with dry teat cleaning could be a better hygiene standard in general. Inadequate farm cleaning has repeatedly been identified as a risk factor for increased ESBL-*E. coli* prevalence [16,44]. Moreover, the practice of wet teat cleaning has the potential for promoting bacterial transmission due to contaminated cleaning solutions, which could result in higher ESBL/AmpC-*E. coli* pressure within the dairy herd [40].

Surprisingly, disinfection of the calving area increases the occurrence of ESBL/AmpC-*E. coli* in fresh cows, although the use of biocides is known to be a reliable method for the reduction of pathogenic bacteria [45]. Unfortunately, the questionnaire did not record the frequency of cleaning and disinfection in the calving area. One possible explanation for the increased prevalence in the dams could be the use of the disinfection measure to compensate for inadequate and reduced cleaning frequency of the calving pen. It is also conceivable that frequent disinfection leads to ESBL/AmpC-*E. coli* developing decreased sensitivity to biocidal products, as this has been shown to occur for other bacteria [46]. Biocide resistance can cause selection advantages, which result in an increased growth of ESBL/AmpC-*E. coli*, leading to increased colonization rates of the dams. Considering that both the daily cleaning of the calf’s feeding equipment and the disinfection of the calving area was associated with high ESBL/AmpC-*E. coli* prevalence in calves and in dams, respectively, suggest that inaccurate cleaning and disinfection procedures carry the risk of worsening the prevalence of ESBL/AmpC-*E. coli*.

Finally, self-production of basic feeds increased the ESBL/AmpC-*E. coli* prevalence in the dams. In each examined farm, the feed ration of the dairy herd was based on corn and grass silage. More than half of the participants self-produced the silages on their own farmland using manure from the calves and dairy cows as organic fertilizers. This may lead to the steady circulation of ESBL/AmpC-*E. coli* between livestock and farmland, as microbial contamination of both grassland and cattle feed has already been proven in the past [47,48]. The constant re-introduction of resistant bacteria via the self-produced feed rations could thus cause increased maternal colonization rates.

## 5. Conclusions

In the present study, almost all cows were dried off with antibiotics and the percentage of farms treating every case of clinical mastitis was high. Therefore, the accumulated amount of WM due to medical treatment was high. Since nearly 70% of the calves examined in the study were fed with WM containing antimicrobial residues, which has been proven to be a risk for the occurrence of ESBL/AmpC-*E. coli* in dairy calves, the WM feeding of calves should be legally banned and alternative feeding options must be put into practice. Moreover, new concepts of antibiotic dry cow therapy (selective dry cow therapy), research data on treating mastitis (high self-recovery) and improved concepts for the safe elimination or bioremediation of WM should be implemented on dairy farms to minimize the spread of AMR into the environment. Beyond that, more attention should be paid to strategies that improve farm hygiene, as it remains a simple method to prevent the transmission of ESBL-*E. coli* among dairy livestock. The ultimate purpose of adjusting the management aspects identified in this study is not only to limit or prevent the spread of ESBL/AmpC-*E. coli*, but to improve overall animal health.

## Figures and Tables

**Figure 1 microorganisms-09-02135-f001:**
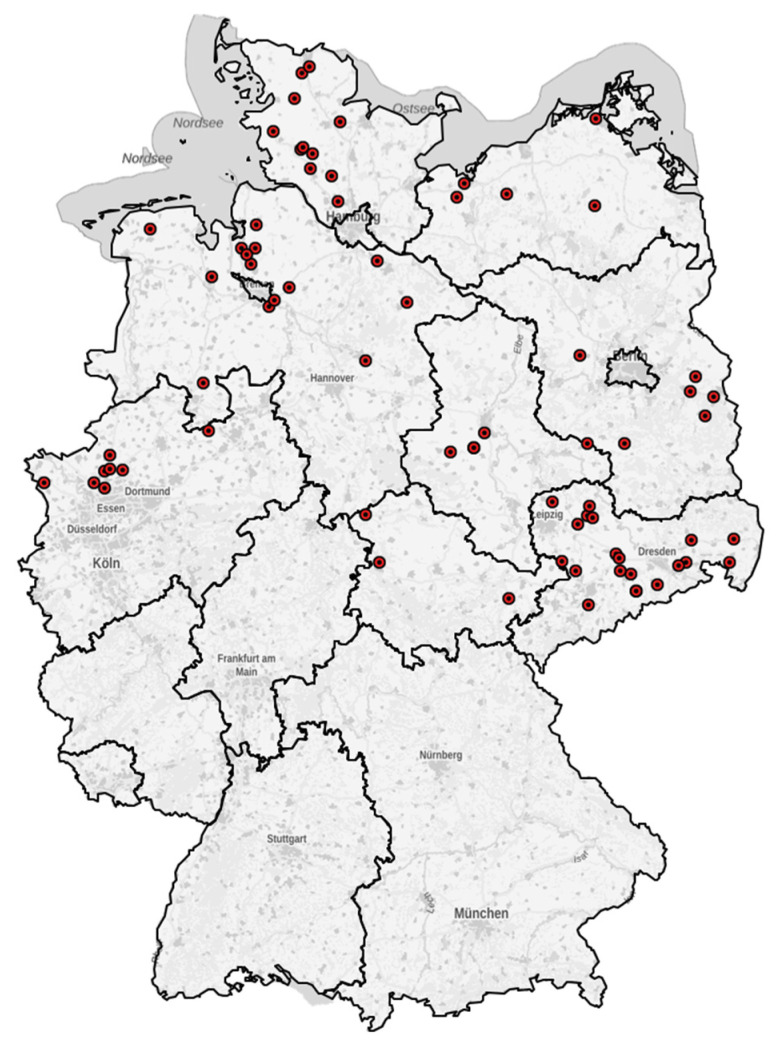
Map of Germany with the localizations of the farms that participated in the study. Each point (•) represents one farm.

**Table 1 microorganisms-09-02135-t001:** Farm prevalence of ESBL/AmpC-*E. coli* in calves, their dams and cow-calf pairs.

	Farm Prevalence of ESBL/AmpC-*E. coli* (%)		
	Calves (*n* = 1442)	Dams (*n* = 1374)	Cow-Calf Pairs (*n* = 1385 *)
Minimum	0	0	0
Maximum	100	88.9	87.5
Mean	63.5	18	13.6
95% confidence interval	57.4–69.5	12.5–23.5	8.7–18.4
Standard deviation	25.6	23.3	20.3
Median	66.1	8.3	5.1
1.Quartile	45.7	0	0
3. Quartile	86.2	28.1	20.6

* Eleven dams gave birth to twins; therefore, the number of calves sampled is higher than the number of dams.

**Table 2 microorganisms-09-02135-t002:** Percentage of farms implementing the studied farm management practices.

Farm Management Factors	Implementation *n* = 72 (%)
Application of ß-lactam antibiotics	100
Antibiotic dry cow therapy in general	91.7
Use of intramammary seal	84.7
Included preventive treatments for cows *	69.4
Existing biogas reactor	66.7
Strictly used treatment schedules for calves	61.1
Self-production of basic feed	55.6
Application of chinolons	55.6
Antibiotic treatment of every case of clinical mastitis	50
Daily cleaning of calf feeding equipment	29.2
Dry teat cleaning before milking process	20.6
Use of disinfection in the calving area	13.9
Sampled calves were treated with antibiotics **	10.0

* Treatments to prevent hypocalcemia, hypoglycemia and digestive disorders, as well as vitamin, salt, and trace element supplementation; vaccination; use of alternative medicines; and management-related procedures such as watering immediately after parturition, removing the placenta for postpartum retention, or measuring internal body temperature during the first 10 days after calving, but not anti-inflammatory or antibiotic therapy. ** antibiotics of treated calves included amoxicillin, cefquinome, enrofloxacin, florfenicol, marbofloxacin, paromomycin, penicillin, streptomycin, sulfadimidine, trimethoprim, oxytetracycline and tulathromycin.

**Table 3 microorganisms-09-02135-t003:** Percentages of farms implementing selected common practices regarding calf feeding.

Calf Feeding Management Factors	Implementation *n* = 72 (%)
Colostrum feeding until three hours after birth	81.9
Identical feeding irrelevant of the sex	80.6
Feeding of waste milk (in total and mixed ration)	66.7
Feeding < 5 L per day	61.1
First meal ≥ 4 L colostrum	45.8
Feeding milk replacer exclusively	19.4

**Table 4 microorganisms-09-02135-t004:** Results of the association between the parameters and the level of ESBL/AmpC-*E. coli* calf prevalence according to the univariable logistic regression model.

Management Factors	Odds Ratio	*p* Value
Use of waste milk (nonsalable)	3.2154	0.0313
In-house biogas reactor	0.2381	0.0363
Treatment schedules for calves	0.2762	0.0392
Application of chinolons	0.3005	0.0395
Using milk replacer exclusively	0.3182	0.0632
Included preventive treatments for cows	0.2807	0.0646
Daily cleaning of calf feeding equipment	2.9000	0.0837
Outsourced heifer rearing	0.3056	0.0853
Calf < 1 h with the dam	0.4162	0.1032
Outsourced heifer rearing at several other locations	0.3478	0.1291

**Table 5 microorganisms-09-02135-t005:** Statistics on management factors influencing the level of ESBL/AmpC-*E. coli* prevalence in calves according to the multivariable logistic regression model.

Management Factors	Odds Ratio	*p* Value
Use of waste milk (nonsalable)	9.65	0.005
Included preventive treatments for cows	0.13	0.029
Daily cleaning of calf feeding equipment	6.03	0.021

**Table 6 microorganisms-09-02135-t006:** Results of the association between the parameters and the level of the maternal ESBL/AmpC-*E. coli* prevalence according to the univariable logistic regression model.

Management Factors	Odds Ratio	*p* Value
Dry teat cleaning	0.2125	0.0124
No disinfection of calving area	0.2987	0.0282
Self-production of basic feed	2.8121	0.0404
Outsourced heifer rearing in another location of the own farm	2.7329	0.0582
Separate husbandry of waste milk cows	2.7143	0.0849
Waste milk cows milked at the end	2.2857	0.1026
Use of Benestermycin^®^	0.4402	0.1120
Co-husbandry of calving and sick cows	3.5714	0.1195
Outsourced heifer rearing at several locations	0.3415	0.1243
Use of lime in the calving area	2.5968	0.1300
Use of Fenicols	2.1685	0.1354
Co-calving on deep bedding	5.7391	0.1397

## Data Availability

Not applicable. Standard practices of animal care and use were applied to animals used in this project. All animal sampling was performed within the guidelines of the German Animal Protection Law, meaning no additional license was required.

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
