# Peer review of "Prevalence and Risk Factors for ESBL/AmpC-E. coli in Pre-Weaned Dairy Calves on Dairy Farms in Germany"

_microorganisms, 2021, doi:10.3390/microorganisms9102135_

Round 1

Reviewer 1 Report

The manuscript "Prevalence and risk factors for ESBL/AmpC-E. coli in pre-weaned dairy calves on dairy farms in Germany" by Weber et al. describes an interesting approach to the investigation of ESBL/AmpC-E. coli through the use of an extensive questionnaire, that considered quite all the critical points of dairy farming, and analysing the relations between these risk factors and the presence of ESBL/AmpC-E. coli.

However, I think that there are a few points that should be reviewed. A wide description of sampling and of numbers of samples should be added (e.g. milk tank samples). Moreover, another criticism is in the isolation and characterization paragraph, because after the culture (and subculture) on CHROM ID agar plates, no other test has been done, or has not been reported, to confirm (i) that the colonies are effectively E. coli and (ii) that they are indeed ESBL (i.g. through a PCR for AMR genes rather than with the ESBL-AmpC-Detection Test or via MIC or diffusion disc). Please see the list below.

Major revisions

  • The manuscript should be revised by an English native speaker.

Materials and Methods

  • Line82: “client-base of Alta Deutschland GmbH were included” is this why no farms in the South of Germany were included? The sampling of farms appears unbalanced towards the North of Germany.
  • Line 93: “Fecal swabs were taken from calves between seven and 28 days of age and their cor-responding dams.” How many sampled calves? How many farms were sampled? I know that all this information are added in the Results paragraph but I think it would be more clear to describe the whole sampling in the Materials and methods paragraph.
  • Lines 99-100: “As only the dams were sampled, cow sampling referred to the calves, which were sampled.” I think I have understood the meaning of this sentence but it should be rephrased because, as it is, it is not easy to understand.
  • Line 105 and Line 116: how many fecal samples? How many milk samples? These numbers should be added. Did Authors sample a fecal swab for all the included individuals, calves and dams? For each sampled farm, was a milk sample taken from the tank?
  • Line 112: are there information available on individual treatments of calves or their dams? I mean, maybe the subjects from which ESBL/AmpC- coli were isolated were treated during their first days of life/or their dams.
  • Line 114: “inclusion of preventive treatment for cows” Does this item mean that the dams “mothers of the calves” had been treated during gestation or before delivery? How they were treated? With antibiotics or anti-inflammatory drugs? This point should be further explained.
  • Lines 116-123: a criticism is related to the lack of any other test, in addition to the use of CHROM ID agar plates, that actually confirms first the detection of coli and second that the colonies isolated were actually ESBL/AmpC.
  • Line 126-128: which was the dependent variable of the Binomial models? The prevalence of ESBL/AmpC- coli of calves or the one recorded for dams? Or which? Which were the binomial parameters included? A wider description of the models should be added. This would make everything, even the subsequent discussion of the results, much clear.

Results

  • Line 153: general comment: many statistical tests have been done and several relationships emerged, thus I think it is necessary to better specify what kind of effect a factor had on the dependent variable. i.e., instead of talking about “positive or protective” effect or “negative” effect, specify that the occurrence/presence of that risk factor increases/decreases the prevalence of ESBL/AmpC- coli. This would help the understanding of the results and to better follow the manuscript.
  • Line 159: “1,442 of them were calves and 1,374 dams.” Why lower number of dams were sampled? This concept should be moved to the Materials and Methods paragraph better describing how the sampling was carried out.
  • Line 160: what Authors mean with “arbitrarily”? Again, a better explanation of the sampling should be included.
  • Line 166: what about the prevalence of ESBL/AmpC- coli in the milk bulk tanks?
  • Line 170: Have the maximum and minimum values of the prevalence of calves and dams been calculated on the basis of the individual farms results? The total number of the analyzed subjects (calves and dams) and the number of positive subjects should be added in the caption or in the Table in order to make these points more clear (e.g. calves (n =…..), their dams (n = …..) and cow-calf pairs (n = ….).
  • Lines 194-196: “The feeding of WM to calves revealed the strongest negative influence on the prevalence of ESBL/AmpC- coli in calves (OR = 9.65; P = 0.005), whereas the inclusion of preventive treatment for cows had a significant protective influence (OR = 196 0.13; P = 0.029) (Table 5).” does this sentence mean that calves fed with WM had a higher prevalence of ESBL/AmpC-E. coli while calves that were born from treated mothers (“inclusion of preventive treatment for cows”) were less likely to have ESBL/AmpC-E. coli?

Discussion

  • Lines 224-228: I think that the concept about the difficulties in comparing results and prevalences emerged using different diagnostic methods, from different studies, should be discussed.
  • Lines 230-232: “Due to convenient sampling, there could be some bias regarding the estimation of prevalence in our study. However, in a quarter of the herds (n = 18) all present calves were sampled (Supplement 3).” Again, a better explanation of the sampling should be added because it is difficult to completely understand the study design and the related limitations.
  • Lines 281-296: this part should be rephrased in order to make it more clear. First, because it is not clear what kind of treatment dairy cows received, whether with antibiotics or anti-inflammatory drugs. Second, it is not clear to me why "the interpretation of this parameter is complicated" and how the Authors said that the treatment seems to have “improved the fitness of the mother”. I think it makes sense that calves of mothers that received treatments showed higher prevalence of ESBL/AmpC- coli.
  • Lines 306-308: “The positive relationship between culling rates and ESBL/AmpC- coli prevalence of the dams underlines the association between better health and lower occurrence of multi-resistant bacteria.” I don’t understand how the culling rates could be positively associated with the prevalence of ESBL/AmpC-E. coli and in any case I don’t even understand the reason that has been included to motivate this result. What do Authors mean with “culling rates”?
  • Lines 309-316: these sentences should be rephrased because the description of the results and their discussion appear confused.
  • Line 317-322: I can't understand in which part of the results these ones are shown. Does this refer to the item “No disinfection of calving area” of Table 6? Anyway, this part appears confused and the hypothesis proposed for discussing results seem a bit forced.

Conclusion

  • Lines 337-354: Some critical management issues emerged from the questionnaire and I think that in this section should be stressed the fact that these bad practices must absolutely be addressed not only to limit/avoid the spread of ESBL/AmpC- coli but also for a better management of farms in general.

Minor revisions

  • Line 148-150: the “ Location of the farms” paragraph and the Figure 1 (Map of the sampled farms) should be moved to the beginning of the Materials and Methods.
  • Line 170, 190, 200, 209: the word “ coli” should be written in italics.

Author Response

The manuscript "Prevalence and risk factors for ESBL/AmpC-E. coli in pre-weaned dairy calves on dairy farms in Germany" by Weber et al. describes an interesting approach to the investigation of ESBL/AmpC-E. coli through the use of an extensive questionnaire, that considered quite all the critical points of dairy farming, and analysing the relations between these risk factors and the presence of ESBL/AmpC-E. coli.

We are really glad that you found merit in our study.

However, I think that there are a few points that should be reviewed. A wide description of sampling and of numbers of samples should be added (e.g. milk tank samples). Moreover, another criticism is in the isolation and characterization paragraph, because after the culture (and subculture) on CHROM ID agar plates, no other test has been done, or has not been reported, to confirm (i) that the colonies are effectively E. coli and (ii) that they are indeed ESBL (i.g. through a PCR for AMR genes rather than with the ESBL-AmpC-Detection Test or via MIC or diffusion disc). Please see the list below.

Thank you for the comments. We hope that we fulfilled all points raised by you. Detailed responses to your concerns are listed below.

Materials and Methods

  • Line82: “client-base of Alta Deutschland GmbH were included” is this why no farms in the South of Germany were included? The sampling of farms appears unbalanced towards the North of Germany.
  • You are right, the sampling is unbalanced towards the north of Germany. However, this is an implication of the study design. As indicated, the present study aimed to analyze the prevalence of this bacterial species on a bigger scale in large German dairy farms. In Germany, we have a north-south divide regarding cow numbers in dairy herds. In the north of Germany, we have large dairies with more than 100 cows and in the south the herds are smaller. To date there is one study for Germany dealing with small herds in the south of Germany (Schmid et al., 2013). Therefore, we wanted to study the prevalence of ESBL/AmpC-coli in larger herds. We added this aspect in the discussion section (Line 255-258)
  • Line 93: “Fecal swabs were taken from calves between seven and 28 days of age and their corresponding dams.” How many sampled calves? How many farms were sampled? I know that all this information are added in the Results paragraph but I think it would be more clear to describe the whole sampling in the Materials and methods paragraph.
  • Sorry for this inconvenience. We now present the data in the Materials and methods section (Line 108).
  • Lines 99-100: “As only the dams were sampled, cow sampling referred to the calves, which were sampled.” I think I have understood the meaning of this sentence but it should be rephrased because, as it is, it is not easy to understand.
  • We rephrased the sentence (Line 130).
  • Line 105 and Line 116: how many fecal samples? How many milk samples? These numbers should be added. Did Authors sample a fecal swab for all the included individuals, calves and dams? For each sampled farm, was a milk sample taken from the tank?
  • We now present the numbers in the Materials and methods section. Yes, we sampled a swap for all included individuals (= 2,816) and we took a tank milk sample from every farm (n= 72) (Line 107-113).
  • Line 112: are there information available on individual treatments of calves or their dams? I mean, maybe the subjects from which ESBL/AmpC-coli were isolated were treated during their first days of life/or their dams.
  • Yes, we recorded the individual treatments of the calves. Ten percent of the calves were treated with antibiotics. We now present this information in the Result section (Table 2). However, antimicrobial treatment was not associated with ESBL/AmpC detection. Most of the cows were dried off with antibiotics, i.e. nearly every cow was pre-treated before sampling.
  • Line 114: “inclusion of preventive treatment for cows” Does this item mean that the dams “mothers of the calves” had been treated during gestation or before delivery? How they were treated? With antibiotics or anti-inflammatory drugs? This point should be further explained.
  • Preventive treatment for cows were treatments to prevent hypocalcemia (e.g. feeding practice, oral calcium substitution, parenteral calcium substitution), hypoglycemia (e.g. oral/ parenteral substitution), indigestion (e.g. rumen stimulants), as well as the substitution of vitamins, salts and trace elements, the implementation of vaccinations (e.g. clostridia, pasteurella, mannheimia, moraxella, etc.), the use of alternative medicinal products (e.g. herbal, homeopathic or enzymatic treatment) and management related procedures like drench of water immediately after delivery, removing the placenta for postpartum retention or measuring the internal body temperature in the first 10 days after calving but does not include anti-inflammatory or antibiotic therapy. Moreover, preventive antibiotic drug delivery is not allowed in Germany and in the whole European Union. We added information on preventive treatments to table 2 (Line 264-269).
  • Lines 116-123: a criticism is related to the lack of any other test, in addition to the use of CHROM ID agar plates, that actually confirms first the detection of coliand second that the colonies isolated were actually ESBL/AmpC.
  • Thank you for this comment. You are right; however, no biochemical, molecular or mass spec analyses were included due to the large number of samples and positive colonies. Moreover, we did not perform these analyses for the following reasons:
  1. CHROM ID orientation agar plates are suitable to identify coli (s. CHROMagar Orientation Pre-poured plates (mast-group.com).
  2. When adding 2 µg/ml cefotaxime, all coli colonies growing on the agar plate are either ESBL- or AmpC-positive. Therefore, we defined and described the bacteria as ESBL/AmpC producers. Due to the large number of isolates and because this is not the focus of our study, we did not determine which genes are responsible for the resistance against cefotaxime.

Please also note that we have previously verified the applied methodology by whole-genome genome sequencing and mass spectrometry in several studies (Homeier-Bachmann et al., 2021; Eger et al., 2021; Schierack et al., 2020). To emphasize this, we cited the mentioned studies in the Materials and Methods section (Line 183-184).

  • Line 126-128: which was the dependent variable of the Binomial models? The prevalence of ESBL/AmpC-coli of calves or the one recorded for dams? Or which? Which were the binomial parameters included? A wider description of the models should be added. This would make everything, even the subsequent discussion of the results, much clear.
  • The dependent variable was the prevalence of calves for the calf model and the prevalence of the dams for the cow model. All parameters of the questionnaire were included as binomial parameters in the logistic regression. The complete questionnaire is available as supplementary data. We added some lines to clarify this (Lines 143-146, 157).

Results

  • Line 153: general comment: many statistical tests have been done and several relationships emerged, thus I think it is necessary to better specify what kind of effect a factor had on the dependent variable. i.e., instead of talking about “positive or protective” effect or “negative” effect, specify that the occurrence/presence of that risk factor increases/decreases the prevalence of ESBL/AmpC-coli. This would help the understanding of the results and to better follow the manuscript.
  • We now described the relationship between prevalence and independent variables as suggested throughout the manuscript.
  • Line 159: “1,442 of them were calves and 1,374 dams.” Why lower number of dams were sampled? This concept should be moved to the Materials and Methods paragraph better describing how the sampling was carried out.
  • We apologize for the inaccuracy of the description. We included this in the Materials and Methods section and describe the sampling in more detail. Furthermore, we added that twins and death of the dams resulted in lower numbers of cows compared to calves (Line 107-113).
  • Line 160: what Authors mean with “arbitrarily”? Again, a better explanation of the sampling should be included.
  • We apologize and changed the explanation of sampling (Line 102-104).
  • Line 166: what about the prevalence of ESBL/AmpC-coli in the milk bulk tanks?
  • We apologize that we missed to state the results of tank milk testing. No ESBL/AmpC-coli were detected in tank milk samples. We add a sentence to the Result section (Line 177)
  • Line 170: Have the maximum and minimum values of the prevalence of calves and dams been calculated on the basis of the individual farms results? The total number of the analyzed subjects (calves and dams) and the number of positive subjects should be added in the caption or in the Table in order to make these points more clear (e.g. calves (n = 1,442), their dams (n = 1,374) and cow-calf pairs (n = 1,385).
  • Yes, the maximum and minimum values have been calculated on the basis of the individual farm results. The requested numbers were added accordingly.
  • Lines 194-196: “The feeding of WM to calves revealed the strongest negative influence on the prevalence of ESBL/AmpC-coli in calves (OR = 9.65; P = 0.005), whereas the inclusion of preventive treatment for cows had a significant protective influence (OR = 196 0.13; P = 0.029) (Table 5).” does this sentence mean that calves fed with WM had a higher prevalence of ESBL/AmpC- coli while calves that were born from treated mothers (“inclusion of preventive treatment for cows”) were less likely to have ESBL/AmpC-E. coli?
  • Yes, this is right. We rewrote the sentence (Line.212-215).

Discussion

  • Lines 224-228: I think that the concept about the difficulties in comparing results and prevalences emerged using different diagnostic methods, from different studies, should be discussed.
  • Thank you for your suggestion. We have included this point in the discussion (Line 247-250).
  • Lines 230-232: “Due to convenient sampling, there could be some bias regarding the estimation of prevalence in our study. However, in a quarter of the herds (n = 18) all present calves were sampled (Supplement 3).” Again, a better explanation of the sampling should be added because it is difficult to completely understand the study design and the related limitations.
  • We now describe the sampling in more detail (s. answers above) However, we clearly mention in the Materials and Methods section that the sample size was calculated according to the calves between seven and 28 days of age which were present at the day of sampling estimating a prevalence of 10%, a confidence interval of 95%, and a desired precision of 5%. To clarify the sample size, we have supplemented the supplementary Table 3 accordingly.
  • Lines 281-296: this part should be rephrased in order to make it more clear. First, because it is not clear what kind of treatment dairy cows received, whether with antibiotics or anti-inflammatory drugs. Second, it is not clear to me why "the interpretation of this parameter is complicated" and how the Authors said that the treatment seems to have “improved the fitness of the mother”. I think it makes sense that calves of mothers that received treatments showed higher prevalence of ESBL/AmpC-coli.
  • Yes, you are right. We clarified accordingly and describe the preventive treatments for cows (s. also above). Furthermore, we added that preventive treatments do not include anti-inflammatory or antibiotic therapy. (Line 192, 311-315).
  • Lines 306-308: “The positive relationship between culling rates and ESBL/AmpC-coli prevalence of the dams underlines the association between better health and lower occurrence of multi-resistant bacteria.” I don’t understand how the culling rates could be positively associated with the prevalence of ESBL/AmpC- coli and in any case I don’t even understand the reason that has been included to motivate this result. What do Authors mean with “culling rates”?
  • Culling rate is the percentage of animals that leave the herd for different reasons (disease, infertility, low milk yield...). This is a general classification number for dairy herds: Usually, the higher the culling rate the lower the general health of the herd. We included the explanation in the manuscript (Line 233-234).
  • Lines 309-316: these sentences should be rephrased because the description of the results and their discussion appear confused.
  • We rephrased the paragraph to make it clearer (Line 338-345).
  • Line 317-322: I can't understand in which part of the results these ones are shown. Does this refer to the item “No disinfection of calving area” of Table 6? Anyway, this part appears confused and the hypothesis proposed for discussing results seem a bit forced.
  • Sorry for the inconvenience. We rephrased the paragraph to make it clearer (Line 346-355).

Conclusion

  • Lines 337-354: Some critical management issues emerged from the questionnaire and I think that in this section should be stressed the fact that these bad practices must absolutely be addressed not only to limit/avoid the spread of ESBL/AmpC-coli but also for a better management of farms in general.
  • Thank you for the suggestion to be a little clearer here. We have amended the conclusion section accordingly (Line 376-378).

Minor revisions

  • Line 148-150: the “ Location of the farms” paragraph and the Figure 1 (Map of the sampled farms) should be moved to the beginning of the Materials and Methods.
  • We moved the paragraph and Figure 1 to the Materials and Methods section as suggested.
  • Line 170, 190, 200, 209: the word “coli” should be written in italics.
  • We write E. coli in italics now.

Reviewer 2 Report

Authors have addressed my previous critiques or have explained issues I raised.   I am satisfied with the updated version.

Author Response

Thank you.

Round 2

Reviewer 1 Report

The manuscript "Prevalence and risk factors for ESBL/AmpC-E. coli in pre-weaned dairy calves on dairy farms in German" by Weber et al. is significantly improved after the revision phase. However, I think that a few points are still to be revised especially concerning the use of the only chromogenic medium plates as the only diagnostic test. Please see the details below.

Major revisions

  • The manuscript should be revised by an English native speaker.

Introduction

  • “Usually, coli from cattle feces carry fewer antibiotic resistances than feces from other livestock animals.” a reference for this sentence should be added.

Materials and Methods

  • CHROM ID orientation agar plates: I think that the values of sensitivity and specificity of this medium and some further specifications to argue for its sole use should be included in the text. I understand the intrinsic difficulties of the analyses of so many samples, and I think this aspect should be included in the text motivating that a confirmatory test, whether it is microbiological or molecular, was not conducted because the use of these plates was sufficient to have a reliable result. Concerning the included three references, I think they are not so useful in this context because in this papers Authors used the chromogenic medium followed by other very solid tests. So these citations, in my opinion, are not so useful to argument what Authors have done in this study.

Results

  • Table 1: “Prevalence cow-calf pairs (%) (n=1,385)” how was it possible that cow-calf pairs number was higher than the one of analysed dams?
  • “ESBL/AmpC- coli could not be detected in any of the tank milk samples.” did all the milk samples resulted negative? As it is written seems that there were some diagnostic problems in the isolation.
  • Table 2: “Sampled calves were treated with antibiotics” which antibiotics were they treated with? any molecules that could have induced resistances to ß-lactams.
  • “In addition, the implementation of daily cleaning of the calf feeding equipment (OR = 6.03; P = 0.021) increased calf prevalence of ESBL/AmpC-coli, as well (Table 5).” and “Surprisingly, disinfection of the calving area increases the occurrence of ESBL/AmpC-E. coli in fresh cows, although the use of biocides is known to be a reliable method for the reduction of pathogenic bacteria”: these two points may appear surprising and unexpected but I think there could also be another explanation, namely the fact that both these effects have been recorded could not be accidental and thus, in some way, the two results could "support" each other. I mean, maybe the farms where farmers clean the “calf feeding equipment” are the same as the farms where farmers disinfect “the calving area” but in both cases farmers do them inaccurately maybe even risking doing worse.

Author Response

The manuscript "Prevalence and risk factors for ESBL/AmpC-E. coli in pre-weaned dairy calves on dairy farms in German" by Weber et al. is significantly improved after the revision phase. However, I think that a few points are still to be revised especially concerning the use of the only chromogenic medium plates as the only diagnostic test. Please see the details below.

Thank you.

Major revisions

  • The manuscript should be revised by an English native speaker.

The manuscript was revised again by a native speaker.

Introduction

  • “Usually, colifrom cattle feces carry fewer antibiotic resistances than feces from other livestock animals.” a reference for this sentence should be added.
  • We now provide a reference for this sentence (Line 61).

Materials and Methods

  • CHROM ID orientation agar plates: I think that the values of sensitivity and specificity of this medium and some further specifications to argue for its sole use should be included in the text. I understand the intrinsic difficulties of the analyses of so many samples, and I think this aspect should be included in the text motivating that a confirmatory test, whether it is microbiological or molecular, was not conducted because the use of these plates was sufficient to have a reliable result. Concerning the included three references, I think they are not so useful in this context because in this papers Authors used the chromogenic medium followed by other very solid tests. So these citations, in my opinion, are not so useful to argument what Authors have done in this study.
  • Thank you very much for your advice. We rephrased this section and describe now in more detail the suitability of the CHROM agar plates and the cefotaxime supplement for our purposes (Lines 136-144). In addition, we explain that due to the high number of isolates obtained in this study and the focus of the study, that we have not further differentiated the isolates (Lines 148-150). As suggested, we removed the citations previously included in this section.

As an example, in the meantime, we performed CTX-M specific RT-PCR (Roschanski et al., 2014) for 45 ESBL/AmpC suspicious isolates from 45 farms. All isolates were CTX-M positive. This confirms Vinueza-Burgos et al. (whom we cite in lines 138-143) as well as our method. However, for the reasons stated above, we cannot perform this assay for more than 1,100 isolates.

Results

  • Table 1: “Prevalence cow-calf pairs (%) (n=1,385)” how was it possible that cow-calf pairs number was higher than the one of analysed dams?
  • We had eleven cows which gave birth to twins. Therefore, we had eleven more cow-calf pairs than analyzed dams. We added this information to Table 1.
  • “ESBL/AmpC-coli could not be detected in any of the tank milk samples.” did all the milk samples resulted negative? As it is written seems that there were some diagnostic problems in the isolation.
  • No, we did not have a diagnostic problem. We rephrased the sentence to make it clearer (Line 189).
  • Table 2: “Sampled calves were treated with antibiotics” which antibiotics were they treated with? any molecules that could have induced resistances to ß-lactams.
  • Thank you for your suggestion. We added this information to Table 2.
  • “In addition, the implementation of daily cleaning of the calf feeding equipment (OR = 6.03; P = 0.021) increased calf prevalence of ESBL/AmpC-coli, as well (Table 5).” and “Surprisingly, disinfection of the calving area increases the occurrence of ESBL/AmpC- coliin fresh cows, although the use of biocides is known to be a reliable method for the reduction of pathogenic bacteria”: these two points may appear surprising and unexpected but I think there could also be another explanation, namely the fact that both these effects have been recorded could not be accidental and thus, in some way, the two results could "support" each other. I mean, maybe the farms where farmers clean the “calf feeding equipment” are the same as the farms where farmers disinfect “the calving area” but in both cases farmers do them inaccurately maybe even risking doing worse.
  • Yes, you are right. We included this explanation in the Discussion section (Line 370-374).